# Impact of Acrylic and Silicone-Based Soft-Liner Materials on Biting Force and Quality of Life of the Complete Denture Wearers: A Randomized Clinical Trial

**DOI:** 10.3390/jcm12052073

**Published:** 2023-03-06

**Authors:** Ahmed Yaseen Alqutaibi, Ahmad A. Alnazzawi, Ahmed E. Farghal, Ramy M. Bakr, Ihab Ismail Mahmoud

**Affiliations:** 1Substitutive Dental Sciences Department, College of Dentistry, Taibah University, Al Madinah 41311, Saudi Arabia; 2Prosthodontics Department, College of Dentistry, Ibb University, Ibb 70270, Yemen; 3Removable Prosthodontics Department, Future University, Cairo 11835, Egypt; 4Removable Prosthodontics, Faculty of Dental Medicine, Al-Azhar University, Cairo 11884, Egypt

**Keywords:** complete denture, soft liner, oral health impact profile, bite force, quality of life, OHRQoL

## Abstract

This rerandomized clinical trial evaluated the influence of soft liners (SL) on biting force, pain perception, and the oral health-related quality of life (OHRQoL) of complete denture wearers. Twenty-eight completely edentulous patients complaining of ill-fitting lower complete dentures were selected to participate in the study from the Dental Hospital, College of Dentistry, Taibah University. All patients received new complete maxillary and mandibular dentures; then they were randomly divided into two groups (14 patients in each group): the acrylic-based SL group, in which the mandibular denture was lined with an acrylic-based soft liner, and the silicone-based SL group, in which the mandibular denture was lined with a silicone-based soft liner. OHRQoL and maximum bite force (MBF) were assessed in this study before denture relining (baseline), then at one month and three months after relining. The finding showed that both treatment modalities significantly improved the OHRQoL of included patients at one-month and three-month periods compared to baseline records (i.e., dentures before relining) with a statistically significant difference (*p* < 0.05). However, there is no statistical difference between groups at the baseline, one-, and three-month follow-up periods. Regarding maximum biting force, when acrylic-based SL is compared to silicone-based SL, there is no statistical difference between groups at baseline (75 ± 31 and 83 ± 32 N) and one-month follow-up periods (145 ± 53 and 156 ± 49 N); however, after three months of function, the silicone-based group recorded 166 ± 57 N statistically significant high biting force compared to the acrylic-based group that recorded 116 ± 47 N (*p* < 0.05). Permanent soft denture liners positively affect maximum biting force, pain perception, and OHRQoL more than conventional dentures. After three months, silicone-based SL outperformed acrylic-based soft liners in maximum biting force, which may indicate better long-term results.

## 1. Introduction

With increased life expectancy, the effective demand and need for complete dentures continue to grow [1]. Nonetheless, wearing complete dentures may have negative consequences for the health of oral and denture-supporting tissues. One of the issues that complete denture wearers face is significant changes in masticatory function compared to dentate subjects. Furthermore, the complete denture wearer experiences more pain in the supporting mucosa and a reduction in the bite force, increasing the risk of masticatory muscle atrophy, which may lead to changes in dietary habits and an impaired nutritional status, resulting in a low-quality life [2]. These patients’ functional difficulties can be reduced using dental implants [3,4] or soft denture liners [5].

Even though dental implants are highly effective in supporting mandibular dentures, as reported in several trials [6,7,8], they are not feasible for all cases because of insufficient underlying bone or the presence of medical and financial restrictions [9]. Alternatively, soft denture liners have several advantages, including no need for surgical intervention and a low treatment cost. Besides, the resilient denture liner materials’ flexibility, resiliency, and shock absorbency help reduce and distribute the masticatory load to the ridge [10,11,12,13].

Denture liners can be either hard [14,15], usually made of polymethylmethacrylate [16,17,18], or soft liner (SL) (i.e., resilient) [19,20,21,22]. Acrylic-based SL is produced by adding plasticizers to the acrylic resin, and silicone-based SL is a mixture of siloxane and silica that remains “soft” after cross-linking/setting [20,22,23]. SL materials have been used in dental fields for over a century as resilient substances attached to the fitting surface of a denture to reduce localized point pressures. Due to their elastic nature, soft liners have a cushion effect as they absorb energy [20], so they can help heal the inflamed mucosa [24,25,26], distribute the functional load in the support area of the prostheses [27,28], and improve their adaptation and retention [29].

Because acrylic-based SL and silicone-based SL have different mechanical properties, their clinical effects differ [30]. In several trials [31,32], the acrylic-based SL’s clinical efficacy and patient satisfaction were compared to conventional dentures. It found that the patients who received dentures with SL were more satisfied with chewing and comfort and had less pain. However, when acrylic-based SL was compared to silicone-based SL, there were conflicting results in the published research on what was better. Some studies [33,34] reported that silicone-based SL was preferred over acrylic ones. Other studies [12,35] revealed that acrylic-based SL is more effective as a denture lining material in terms of durability. Unfortunately, most of the published literature that compared acrylic-based SL and silicone-based SL were in vitro studies or nonrandomized clinical trials.

As a result, solid research based on a well-conducted randomized clinical trial regarding the effect of different permanent soft denture liners is currently lacking. This study aimed to evaluate the impact of acrylic- and silicone-based soft liners on pain perception, maximum biting force (MBF), and oral health-related quality of life (OHRQoL) in complete denture wearers. The research question is: in patients with ill-fitting mandibular complete dentures, will the acrylic-based SL improve OHRQoL, biting force, and pain perception compared to the silicone-based SL?

The null hypothesis was that the improvement in OHRQoL, biting force, and pain perception would be the same for complete denture wearers with acrylic or silicone-based soft liners.

## 2. Materials and Methods

### 2.1. Trial Design

A double-blind, randomized clinical trial was carried out, with design and reporting following the CONSORT guidelines. The research was carried out at Taibah University Dental Hospital in Madinah, a city in Saudi Arabia’s western region.

A signed, written, informed consent was taken from patients before starting any procedures. The Research Ethics Committee at Taibah University approved the proposal for this study (approval # 11012020).

### 2.2. Participants

The study population consisted of those completely edentulous patients complaining of ill-fitting lower complete dentures and wishing to construct new dentures. They were selected to participate in this study. The inclusion criteria include participants being able to follow simple instructions and sign informed consent. They had enough inter-arch distance to accommodate, and with a class I jaw relationship (i.e., the maxillary alveolar ridge crest is directly above the mandibular ridge). On the other hand, bruxers and those with temporomandibular disorders, uncontrolled systemic disease, severe oral manifestations, xerostomia, or psychological or psychiatric disturbances that would affect treatment were excluded.

The size estimation was calculated using the OHRQoL rating as the primary outcome for this trial to determine the appropriate sample size. To fulfill the criteria of 80% power with a two-sided alpha level of 5% and to factor in potential participant dropouts, 24 subjects were enrolled in this study. The difference in means of OHIP between the two groups is set at 22. Based on the mean ± standard deviation of the overall OHIP score obtained in a previous study [36] (silicone-based SL: 25.8 ± 6.2, acrylic-based SL: 32.4 ± 5.9). However, assuming that 10% of participants drop out, the target number of participants is 28.

### 2.3. Interventions

All participants received new complete upper and lower dentures constructed according to a standard protocol, and the dentures were inserted following adequate adjustments. Patients are allowed to use the denture for three months; the baseline records were taken at the end of this period. Patients were randomly divided into two groups (14 patients in each group): the acrylic-based SL group, in which the mandibular denture was lined with an acrylic-based soft liner (VertexTMSoft, Vertex-Dental, Zeist, The Netherlands); and the silicone-based SL group, in which the mandibular denture was lined with a silicone-based soft liner (Molloplast-BTM DETAX). The mandibular dentures were relined following the manufacturer’s recommendations. The soft liner’s thickness was controlled using a vacuum-formed spacer constructed on a mandibular duplicated cast used to create a space of 2 mm for soft liners.

### 2.4. Outcomes

The primary outcome was the oral health-related quality of life (OHRQoL), and the secondary outcomes were the maximum bite force (MBF) and pain perception.

OHRQoL and maximum bite force (MBF) were assessed in this study before denture relining (baseline), then at one month and three months after relining. The pain perception during baseline and the first adjustment session following the initial fitting of the denture.

This study used the Arabic version of the OHIP-EDENT questionnaire [37]. The original questions were grouped according to seven subscales, or domains. Patients responded by rating how frequently oral health problems interfered with their daily activities (4: very often, 3: fairly often, 2: occasionally, 1: rarely, and 0: never). The OHIP-EDENT scores range from 0 to 76, with lower scores indicating better OHRQoL. Care providers were counseled to avoid making comments about treatment possibilities to subjects and were not present when subjects completed the OHIP questionnaires. The pain perception during the first adjustment session was assessed using a 100-mm VAS scale.

The biting force was recorded using an occlusal force meter (GM10; Nagano Keiki, Tokyo, Japan). An occlusal force meter measured the MBF bilaterally at the first molar region. The patient was asked to sit upright, and the instrument was positioned in the first molar region. At maximum intercuspation, the patient was directed to bite as hard as possible three times per side, with a two-minute rest period. The mean maximum biting occlusal force (N) for the three readings was calculated using the patient’s MBF, and the mean of two sides was taken [38]. The MBF was assessed by an independent assessor, unaware of the nature of the interventions.

### 2.5. Randomization

#### 2.5.1. Sequence Generation

The random sequence was generated using a particular website concerned with the randomization process known as Research Randomizer (https://www.randomizer.org/) (accessed on 2 April 2021), where the patients were randomly divided into two groups (14 patients in each group).

#### 2.5.2. Allocation, Concealment Mechanism, and Implementation

One researcher, who was not involved in the patient selection process, was aware of the randomization procedure and had access to the randomization lists saved on a password-protected portable computer. The randomized codes were sealed in sequentially numbered, identical, and opaque envelopes. Patients were asked to pick one of the envelopes, and the investigator, aware of the randomization method, was asked about the particular group and treated appropriately.

### 2.6. Blinding

It was impossible to blind the dentists who made the lined dentures. However, the outcome assessor was blind to the participant’s group. Patients were also kept unaware of any information regarding their assigned group.

### 2.7. Statistical Methods

Statistical analysis was performed using SPSS version 21 (SPSS Inc.). Patients’ baseline characteristics for the two interventions were recorded using frequency distributions and descriptive statistics using a Student’s *t*-test for continuous variables and a chi-square test for categorical data. For the OHRQoL, MBF, and pain perception, mean values were compared by an independent *t*-test to compare the two groups and an ANOVA to compare the effect of time in each group. The significance level was set at 5% for all statistical analyses.

## 3. Results

A sample of 28 patients with a mean age of 59.4 years was included in this study. A total of 28 complete dentures were received.

### 3.1. Participant Flow

All patients (14 in acrylic-based SL and 14 in silicone-based SL) attended the three-month follow-up, as observed in the participants’ flow chart (Figure 1).

### 3.2. Recruitment

Between March and December 2021, the patients were recruited. 28 of the 57 patients investigated met the inclusion criteria. All of them were asked to participate and accepted.

### 3.3. Baseline Data

Table 1 displays the baseline demographic characteristics of the patients who participated in the study. The baseline characteristics of the patients assigned to the treatment groups (acrylic-based SL and silicone-based SL) were similar.

There were no significant variations in any baseline features between the acrylic- and silicone-based soft-liner groups regarding age, edentulous period, the height of the alveolar ridge, mucosal thickness, mucosal resiliency, and denture difficulty classification (*p* > 0.05), as shown in Table 1.

### 3.4. Numbers Analyzed

The data of 28 patients, 14 in the acrylic-based SL group and 14 in the silicone-based SL group, were analyzed.

### 3.5. Outcomes and Estimation

#### 3.5.1. The Oral Health-Related Quality of Life (OHRQoL)

Both treatment modalities significantly improved the quality of life of included patients at one-month and three months periods compared to baseline records (i.e., denture before relining) with a statistically significant difference (*p* < 0.05). However, there is no statistical difference between groups at the baseline, one-, and three-month follow-up periods. The improvement in OHRQoL was slightly more significant in the silicone-based SL group than in the acrylic-based group; nonetheless, this difference was neither statistically nor clinically significant (Table 2).

#### 3.5.2. The Pain Perception

The pain perception during the first adjustment session following the initial fitting of the denture was assessed using a 100 mm VAS scale. There was a significant reduction in pain rating for both SL groups compared to conventional dentures, with no difference between acrylic-based and silicone-based SL. The pain ratings are shown in Table 3.

#### 3.5.3. Maximum Biting Force

Regarding MBF, both treatment modalities improved the biting force of included patients compared to baseline records with a statistically significant difference (*p* < 0.05). When acrylic-based SL is compared to silicone-based SL, there is no statistical difference between groups at baseline (75 ± 31 and 83 ± 32 N) and one-month follow-up periods (145 ± 53 and 156 ± 49 N); however, after three months of function, the silicone-based group recorded 166 ± 57 N of statistically significant high biting force compared to the acrylic-based group that recorded 116 ± 47 N (*p* < 0.05) as shown in Table 4.

## 4. Discussion

This randomized clinical trial evaluated the influence of the lining of lower conventional complete dentures with acrylic versus silicone-based SLs on OHRQoL, pain perception, and MBF. The findings revealed that edentulous individuals who wore complete mandibular dentures with SL had better OHRQoL and MBF and experienced less pain than those with conventional dentures. The bite force and patient quality of life measurements are among the outcomes reflecting complete denture satisfaction and masticatory function.

In this study, both treatment modalities significantly improved OHRQoL compared to conventional complete dentures.

Recent years have seen a shift in the focus of dentistry from valuing only clinical assessments to measuring patients’ subjective experiences. This change in approach has led to the development of an essential construct, OHQoL. The OHIP-EDENT is a shortened version of the OHIP-49, which comprises 19 items targeting edentulous patients. This tool was selected from other questionnaires as it is more suitable for our study and could detect OHRQoL changes in edentulous patients with new or different prostheses [39,40]. The OHIP-EDENT demonstrated satisfactory validity, reliability, and agreement with reported complaints in various languages. It appears valid and reliable for assessing the OHRQoL associated with oral health [41,42,43,44,45,46].

In complete denture wearers, the pain of supporting tissue is a significant concern that affects function and treatment success [47,48]. The results of this study revealed a significant reduction in pain rating for both SL groups compared to conventional dentures, with no difference between acrylic-based and silicone-based SL. The first adjustment session after the initial fitting of the denture was the most opportune moment to establish the effects of soft liners for complete denture users since difficulties impacting the mucosa during this period are possibly the most severe and bothersome for complete denture wearers, as highlighted by Kimoto et al. [47]. In agreement with our findings, Kimoto et al. [47] found that patients with acrylic-based SL experienced less pain and fewer alveolar problems during the first adjustment session than those with conventional dentures.

The relined complete mandibular denture wearers experienced lower physical pain scores compared to baseline measures (i.e., conventional dentures). Although the pain perception measurement in this study is considered subjective, the reduction of pain recorded with VAS is supported by the finding of reduced physical pain scores (OHIP-DENT). This effect could be attributed to SL’s crushing effect, which results in less discomfort and fewer sore spots in the supporting and limiting structures and is more comfortable [39,47]. Based on these responses, patients with soft-relined dentures reported less physical discomfort than those with conventional dentures. According to Beck [48], 63% of mandibular denture difficulties were caused by discomfort. As a result, pain is a crucial concern for denture users. Furthermore, another study [49] suggests that doctors should handle mandibular dentures with caution owing to their better innate responsiveness to external stimuli than the maxilla. Thus, adding SL to mandibular dentures is therapeutically advantageous for edentulous individuals.

Bite force varies depending on where it is applied in the oral cavity. It is most significant in the area of the first molar (i.e., nearly 80% of total bite force is distributed there), and it is easier and faster to measure there. A series of recordings is more reliable than a single recording [50,51]. Both acrylic and silicone-based SL had a higher mean MBF than the conventional denture. This observation was in agreement with Hayakawa et al., who showed that the MBF increased following the placement of a denture with a silicone-based SL [5]. Denture wearers had only one-fifth to one-fourth of the occlusal force and masticatory performance of normal dentition [52]. Complete denture users reach the limit of their occlusal force sooner than those with natural dentition; this might be due to pressure from the denture base on the underlying mucosa and the pain threshold being reached. Thus, the higher MBF in both SLs could be attributed to the liner’s elasticity, which reduced stress on the alveolar ridge mucosa, and the increased pain threshold, which made the alveolar ridge mucosa more resilient to stress [53].

In this study, acrylic- and silicone-based SL had comparable effects on OHRQoL and MBF records, where both groups outperformed conventional dentures. This finding could be interpreted as a result of the more evenly distributed load on the underlying mucosa reported with soft liners, which relieves the supporting structures of excessive mechanical stress [53]. Consequently, the patients experience less pain and ulcers on the ridge, resulting in a longer occluding phase of the masticatory cycle and the ability to apply more force. This can be attributed to resilient liners absorbing energy and preventing force transmission to the underlying tissues [31]. Moreover, soft denture liners used with complete mandibular dentures improved edentulous patients’ masticatory ability. Compared to the conventional hard denture base, soft liners provided patients with fewer problems affecting the alveolar ridge during the first adjustment following the denture insertion [32,54].

There is no statistical difference between groups at one-month follow-up periods when acrylic-based SL is compared to silicone-based SL regarding MBF. However, after three months of function, the silicone-based group recorded a statistically significantly higher biting force than the acrylic-based group. The low molecular weight plasticizer that leaches out into the water could explain this in acrylic materials; at the same time, the water is absorbed into the materials, resulting in viscoelastic property loss, dimensional change, and deterioration of surface conditions. Over time, the silicone products remained stable due to the components’ low water absorption and solubility [33,34].

This finding agreed with a randomized clinical study by Kimoto et al. [32]. They investigated how acrylic-based SL affected complete denture wearers’ masticatory ability and concluded that masticatory efficiency was worse several months after lined denture insertion than at the time of insertion because the material had lost its initial softness.

Murata et al. [12] compared several soft liners to hard resins one week after denture insertion. MBF, patient satisfaction, and chewing time were all clinically assessed. The results revealed that the soft liners surpassed hard dentures in all assessed variables, which agreed with our findings. However, they report better results with acrylic-based SL than with silicone-based SL. The short duration of their study could explain this result, and a more extended observation period could have influenced the results.

The similarity of participants’ baseline characteristics (i.e., age, edentulous period, the height of the alveolar ridge, mucosal thickness, mucosal resiliency, and denture difficulty classification) between the two groups demonstrated that the randomization was adequate. The results of a well-executed randomization demonstrated high internal validity. Thus, unknown interaction variables that affect the results would be controlled. However, this research has shortcomings. First, this study was conducted by a single research facility that did not cover nearly all of Saudi Arabia, resulting in poor external validity. Second, we evaluated MBF, which is considered a static parameter that reflects masticatory efficiency. Individuals who create more forces during mastication have higher masticatory efficiency, so the MBF might be used as a clinical indicator to evaluate masticatory function. A multicenter, well-designed, randomized clinical trial with a broad range of sequentially selected people evaluating dynamic parameters of masticatory efficiency should be conducted to acquire convincing evidence.

## 5. Conclusions

This trial reveals that edentulous individuals wearing complete mandibular dentures with SL had superior biting force and OHRQoL than individuals wearing conventional complete dentures. After three months of function, silicone-based SL outperformed acrylic-based soft liners in maximum biting force, which may indicate better long-term results.

## Figures and Tables

**Figure 1 jcm-12-02073-f001:**
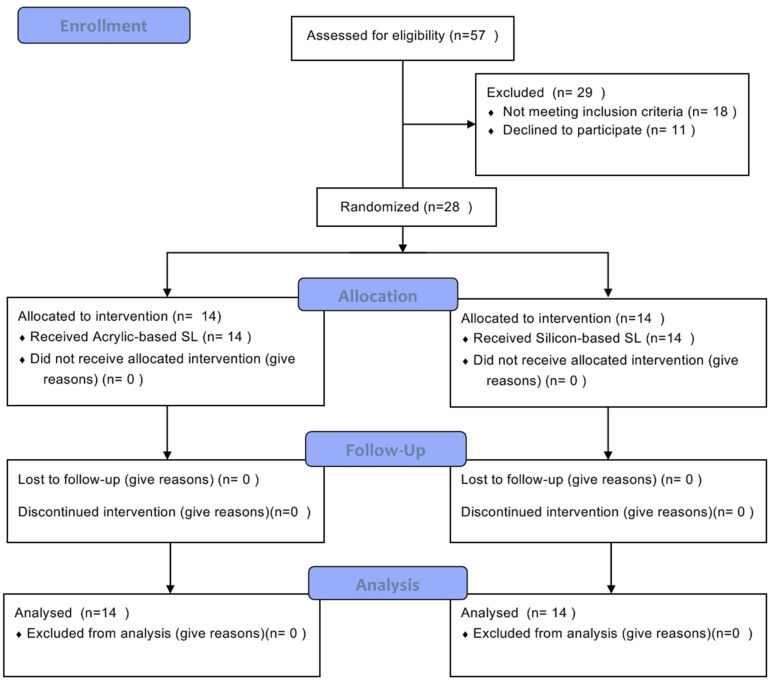
Participants’ flow chart of the study.

**Table 1 jcm-12-02073-t001:** Baseline characteristics of the study participants.

Variable	Acrylic-Based SL Group (*n* = 14)	Silicone-Based SL Group (*n* = 14)
Age (years), mean ± SD	60.1 ± 12.4	58.7 ± 13.1
Edentulous period (mandible) (years), mean ± SD	5.9 ± 2.3	6.4 ± 1.8
Height of alveolar ridge (mm), mean ± SD	16.9 ± 4.2	17.6 ± 3.3
Mucosal thickness (mm), mean ± SD	1.9 ± 0.7	2.2 ± 0.2
Mucosal resiliency (firm, resilient, flappy)	(10, 4, 0)	(9, 5, 0)
Denture difficulty classification (I, II, III, IV)	(6, 7, 1, 0)	(7, 6, 1, 0)

SL, soft liner; SD, standard deviation; mm, millimeter.

**Table 2 jcm-12-02073-t002:** The overall oral health-related quality of life (Mean ± SD) at baseline and one and three months after denture reline.

Interventions	Baseline	One Month	Three Months
Acrylic-Based SL	56.3 ± 8.2 ^Aa^	34.6 ± 7.8 ^Bb^	29.4 ± 5.9 ^Bb^
Silicone-Based SL	53.1 ± 12.8 ^Aa^	31.9 ± 7.4 ^Bb^	25.8 ± 6.2 ^Bb^
MD (CI)	−3.2 (−11.5 to 5.1)	−2.7 (−8.6 to 3.2)	−3.6 (−8.3 to 1.1)

A, B: significant statistical differences can be found in values with dissimilar capital letters in columns (between groups). a, b: significant statistical differences exist between values with dissimilar small letters in rows (between time intervals); MD, mean difference; CI, confidence interval.

**Table 3 jcm-12-02073-t003:** Pain perception (Mean ± SD) rating on the 100-mm VAS.

Interventions	Baseline (Before Relining)	At the Time of the Initial Adjustment
Acrylic-Based SL	63.5 ± 29.8 ^Aa^	47.1 ± 34.9 ^Bb^
Silicone-Based SL	61.1 ± 31.1 ^Aa^	45.9 ± 36.1 ^Bb^
MD (CI)	−2.4 (−26.1 to 21.2)	−1.2 (−28.8 to 26.3)

A, B; significant statistical differences can be found in values with dissimilar capital letters in columns (between groups). a, b; significant statistical differences exist between values with dissimilar small letters in rows (between time intervals); MD, mean difference; CI, confidence interval.

**Table 4 jcm-12-02073-t004:** Maximum bite force in Newtons (Mean ± SD) at baseline; 1, 2, and 3 months after denture adjustment.

**Interventions**	**Baseline**	**One Month**	**Three Months**
Acrylic-Based SL	75 ± 31 ^Aa^	145 ± 53 ^Bb^	116 ± 47 ^Bc^
Silicone-Based SL	83 ± 32 ^Aa^	156 ± 49 ^Bb^	166 ± 57 ^Cb^
MD (CI)	8 (−16.4 to 32.4)	11 (−28.6 to 50.6)	50 (9.4 to 60.5)

A, B, and C: significant statistical differences can be found in values with dissimilar capital letters in columns (between groups); a, b, and c: significant statistical differences exist between values with dissimilar small letters in rows (between time intervals); MD, mean difference; CI, confidence interval.

## Data Availability

Data is available upon request from authors.

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
