# Peer review of "Impact of Acrylic and Silicone-Based Soft-Liner Materials on Biting Force and Quality of Life of the Complete Denture Wearers: A Randomized Clinical Trial"

_jcm, 2023, doi:10.3390/jcm12052073_

Round 1

Reviewer 1 Report

The manuscript by Alqutaibi et al. reports an interesting experimental study in humans, comparing silicone- and acrylic resin-based soft reliners for lower complete dentures. The study is relevant for the field and the manuscript, in general, is clear and well written.

I am sending a marked pdf version of the manuscript with some suggested edits and comments. Other suggestions for improvement of the manuscript follow:

·        Authors use the terms “silicon” and “silicone” interchangeably. Unless you are really mentioning the chemical element only, replace al occurrences of “silicon” by “silicone”.

·        Line 51 (2nd page): This information “…and silicon-based SL is produced by adding plasticizers to 51 the silicone-based elastomers” does not seem accurate. Most silicone reliners do not contain specific plasticizers, but are rather composed by a mixture of siloxane and silica, which remains “soft” after cross-linking/setting.

·        Please explain whether participants, clinical care providers, and/or researchers involved in data collection were blind to received intervention/relining material.

·        The manuscript should define which is the study primary outcome. This will be relevant for my comment on sample size estimation (attached pdf file), which should be based on the primary outcome.

·        The first time you mention the acronym “SL”, please add an explanation of what it means.

·        Please provide the 95% confidence intervals, at least for your main comparisons, i.e., between-group comparisons. This will add important data about the reliability of the study and how certain we could be regarding the non-rejection of a null hypothesis.

Author Response

-The manuscript by Alqutaibi et al. reports an interesting experimental study in humans, comparing silicone- and acrylic resin-based soft reliners for lower complete dentures. The study is relevant for the field and the manuscript, in general, is clear and well written.

-I am sending a marked pdf version of the manuscript with some suggested edits and comments.

Thank you very much for your valuable comments and for taking the time to go through the manuscript, which helped us improve the context and provide the manuscript in a better form for the reader. All your concerns and amendments have been revised accordingly.

-Other suggestions for improvement of the manuscript follow:

  • - Authors use the terms “silicon” and “silicone” interchangeably. Unless you are really mentioning the chemical element only, replace al occurrences of “silicon” by “silicone”.

Thank you, the word silicon was replaced with silicone,  as requested

  • - Line 51 (2nd page): This information “…and silicon-based SL is produced by adding plasticizers to 51 the silicone-based elastomers” does not seem accurate. Most silicone reliners do not contain specific plasticizers, but are rather composed by a mixture of siloxane and silica, which remains “soft” after cross-linking/setting.

 Thank you for your precious note; the text has been amended accordingly.

  • - Please explain whether participants, clinical care providers, and/or researchers involved in data collection were blind to received intervention/relining material.

 Thank you, the following sentence has been added “Both patients and the outcome assessor were blinded to the intervention.”  

  • - The manuscript should define which is the study primary outcome. This will be relevant for my comment on sample size estimation (attached pdf file), which should be based on the primary outcome.

Amended as requested; “The size estimation was calculated using the OHRQoL rating as the primary outcome for this trial to determine the appropriate sample size.”

  • -The first time you mention the acronym “SL”, please add an explanation of what it means.

 Amended as requested

  • -Please provide the 95% confidence intervals, at least for your main comparisons, i.e., between-group comparisons. This will add important data about the reliability of the study and how certain we could be regarding the non-rejection of a null hypothesis.

We appreciate your remarkable suggestion, and the mean differences with confidence interval have been added to tables 2 to 4. 

Reviewer 2 Report

Dear Authors,

The article: 'OImpact of Acrylic-based and Silicon-based soft-liner materials on biting force and quality of life of the complete denture wearers. A Randomized clinical trial' was to evaluate the impact of different acrylic-based and silicon-based soft liners on pain perception, maximum biting force (MBF), and oral health-related quality of life of complete denture wearers.

English language and style are fine.

Punctuation mistakes should be corrected, e.g., p value should be in italics.

Introduction is clearly written.

Figure 1 should be greater. She is illegible.

Add a table with abbreviations.

References should be prepared in accordance with the MDPI guidelines.

To sum up, article should be reconsidered after minor revision.

Author Response

The article: 'Impact of Acrylic-based and Silicon-based soft-liner materials on biting force and quality of life of the complete denture wearers. A Randomized clinical trial' was to evaluate the impact of different acrylic-based and silicon-based soft liners on pain perception, maximum biting force (MBF), and oral health-related quality of life of complete denture wearers.

Response: Thank you very much for your valuable comments and for taking the time to review the manuscript, which helped us improve the context and provide the reader with a better form. All your concerns have been revised accordingly.

English language and style are fine.

Punctuation mistakes should be corrected, e.g., p value should be in italics.

Response: Many thanks for your comment. According to your kind recommendation, the text has been corrected for any language typo and Punctuation problems.

Introduction is clearly written.

Response: Thank you very much.

Figure 1 should be greater.

               Response: Amended as requested.

Add a table with abbreviations.

Response: The abbreviations list added to the end of the manuscript.

References should be prepared in accordance with the MDPI guidelines.

               Response: Amended as requested.

To sum up, article should be reconsidered after minor revision.    

Response: Thank you very much.

Reviewer 3 Report

“This study aimed to evaluate the impact of different acrylic-based and silicon-based soft liners on pain perception, maximum biting force (MBF), and oral health-related quality of life of complete denture wearers.”

General remarks

The topic is interesting and important to warrant publication. The paper is written well, properly organized, and easy to follow. The information is presented in an open-minded and objective manner.

However, the following minor concerns have to be revised before publication.

Minor revisions:

1.      The affiliation section- The e-mails of all authors should be added according to the journal specifications

2.      Abstract- specify what is SL?

3.      Abstract- add to the aim ‘pain perception’.

4.      Abstract-It is not clear to what result this sentence is referring: ‘When acrylic-based compared to silicon-based SL, there is no statistical difference between groups at baseline (75 ± 31and 83 ± 32 N) and one-month follow-up periods (145 ± 53 and 156 ± 49 N’. Therefore, it should be added: Regarding maximum biting force…

5.      Keywords- all in Capital letter.

6.      Introduction- It should be emphasized that permanent SL are being evaluated.

7.      Introduction- add please a ‘null hypothesis’.

8.      The number of references should be before the end of the sentence.

9.      Line 41- ‘these’ should be with capital letter.

10.   Results- Table 1- How you measured the ‘Height of mandibular ridge?’

11.   Line 248- please delete ‘For research articles with several authors, a short paragraph specifying their 248 individual contributions must be provided. The following statements should be used’.

12.   Line 254- Please delete ‘Please add:’.

13.   References- should be written according to the journal specifications.

14.   References 45 and 52 are the same-delete No 52 and please correct in the text.

Author Response

The topic is interesting and important to warrant publication. The paper is written well, properly organized, and easy to follow. The information is presented in an open-minded and objective manner.

However, the following minor concerns have to be revised before publication.

Response: Thank you very much for your valuable comments and for taking the time to review the manuscript, which helped us improve the context and provide the reader with a better form. All your concerns have been revised accordingly.

Minor revisions:

The affiliation section - The e-mails of all authors should be added according to the journal specifications.

Abstract - specify what is SL?

Response: Amended as requested.

Abstract- add to the aim ‘pain perception’.

Response: Amended as requested.

Abstract-It is not clear to what result this sentence is referring: ‘When acrylic-based compared to silicon-based SL, there is no statistical difference between groups at baseline (75 ± 31and 83 ± 32 N) and one-month follow-up periods (145 ± 53 and 156 ± 49 N’. Therefore, it should be added: Regarding maximum biting force…

Response: Amended as requested.

Keywords - all in Capital letter.

Response: Amended as requested.

Introduction - It should be emphasized that permanent SL are being evaluated.

Response: Amended as requested.

Introduction - add please a ‘null hypothesis’.

Response:Thank you very much for your valuable comments. The following hypothesis was added. The null hypothesis was that the improvement in the OHRQoL, biting force, and pain perception would be the same for complete dentures with acrylic or silicone-based soft liners.

The number of references should be before the end of the sentence.

Response: Amended as requested.

C9. Line 41- ‘these’ should be with capital letter.

Response: Amended as requested.

Results - Table 1- How you measured the ‘Height of mandibular ridge?’

Measurements of mandibular bone height were done on the OPG, using the same reference points required for obtaining the measurements as described by the ACP (i.e., The lowest height of the edentulous mandible was measured by recording the distance between superior and inferior borders of the mandible).

Line 248 - please delete ‘For research articles with several authors, a short paragraph specifying their 248 individual contributions must be provided. The following statements should be used’.

Response: Thanks. Amended as requested.

Line 254 - Please delete ‘Please add:’.

Response: Thanks. Amended as requested.

References - should be written according to the journal specifications.

Response: Thanks. Amended as requested.

References 45 and 52 are the same - delete No 52 and please correct in the text.

Response: Thanks. Amended as requested.

Reviewer 4 Report

1. There are so many typographical and grammatical errors. Professional editing is mandatory.

2. L42: [3, 4], These references are related to dental implants only. Those related to resilient denture liners must be cited.

3. Material and Methods: Patients allocation is critical in this study. However, the description of the morphological features of the lower jaw of the two groups are completely lacking. Parameters may include objective measurements such as width and softness of the alveolar crest and movable mucosa area etc. Was there any statistical difference in these parameters in the two group?

4. L69: “Angle class I jaw relationship” How was this determined? Subjects were edentulous so that the anterior alveolar ridge is changed morphologically. Needs practical details for determination.

5. Material and Methods: This study evaluated maximum bite force (MBF) only to represent the masticatory function. The MBF is just a “static” parameter, not dynamic ones such as jaw trajectory and electromyographic activity during chewing etc. It is questionable whether this is enough to represent the masticatory function.

6. Material and Methods: The determination of the pain perception is subjective and easy to be biased.

7. L171-172: It is not understandable why are “bite force and …measurements” preferable outcomes reflecting complete denture service quality.

Author Response

  1. There are so many typographical and grammatical errors. Professional editing is mandatory.

Many thanks for your valuable comment. According to your kind recommendation, professional editing and correction for any language typos were performed.

  1. L42: [3, 4], These references are related to dental implants only. Those related to resilient denture liners must be cited.

Thank you for your precious note; Citation related to resilient denture liners has been added as requested.

  1. Material and Methods: Patients allocation is critical in this study. However, the description of the morphological features of the lower jaw of the two groups are completely lacking. Parameters may include objective measurements such as width and softness of the alveolar crest and movable mucosa area etc. Was there any statistical difference in these parameters in the two group?

Many thanks for your valuable comment. The text and table 1 have been amended to define this concern clearly

“There were no significant variations in any baseline features between the acrylic- and silicone-based soft-liner groups regarding age, edentulous period, the height of the alveolar ridge, mucosal thickness, mucosal resiliency, and denture difficulty classification (P > 0.05), as shown in Table 1.”

  1. L69: “Angle class I jaw relationship”How was this determined? Subjects were edentulous so that the anterior alveolar ridge is changed morphologically. Needs practical details for determination.

Thanks for your valuable comment; we are sorry that was a mistake; we mean skeletal, not angle classification, so the text has been amended.  

Skeletal class I jaw relationship (i.e., The maxillary alveolar ridge crest is directly above the mandibular ridge)

  1. Material and Methods: This study evaluated maximum bite force (MBF) only to represent the masticatory function. The MBF is just a “static” parameter, not dynamic ones such as jaw trajectory and electromyographic activity during chewing etc. It is questionable whether this is enough to represent the masticatory function.

Thanks for your valuable comment; this point has been added to the limitations of this study at the end of the discussion section.

  1. L171-172: It is not understandable why are “bite force and …measurements” preferable outcomes reflecting complete denture service quality.

We appreciate your great comments. The text has been amended to be “ The bite force and patient quality of life measurements are among the outcomes reflecting complete denture satisfaction and masticatory function.”

Round 2

Reviewer 4 Report

Each and every comment noted by the reviewers was addressed, and this is commendable. However, in the nature, the article is very superficial in terms of aim, design, analysis, findings and discussion, which has not been changed at all.

Author Response

  • Each and every comment noted by the reviewers was addressed, and this is commendable. However, in the nature, the article is very superficial in terms of aim, design, analysis, findings and discussion, which has not been changed at all.

Dear reviewer, thank you very much for your valuable comments and for reviewing the manuscript, which helped us improve the context and give the reader a better form. All your concerns and amendments have been revised accordingly.

We reviewed all sections and revised and amended the introduction, method, results, and discussion. 

  • Aim: we rephrase it and add the research question and null hypothesis. 

"This study aimed to evaluate the impact of different acrylic-based and silicon-silicone-based soft liners on pain perception, maximum biting force (MBF), and oral health-related quality of life (OHRQoL) of complete denture wearers. The research question is, in patients with ill-fitting mandibular complete dentures, will the acrylic-based SL improve OHRQoL, biting force, and pain perception compared to sili-cone-based SL? The null hypothesis was that the improvement in the OHRQoL, biting force, and pain perception would be the same for complete denture wearers with acrylic or silicone-based soft liners."

  • Method : The following subheadings were added with elaborate details

2.1 Trial Design

2.2  Participants

2.3 Interventions

2.4  Outcomes

2.5 Randomization

2.5.1 Sequence generation

2.5.2  Allocation concealment mechanism and implementation. 

2.6  Blinding

2.7 Statistical methods

  • Results: The following subheadings were added with elaborate details

3.1 Participant flow

3.2 Recruitment

3.3 Baseline data

3.4 Numbers analyzed.

3.5 Outcomes and estimation

3.5.1 The oral health-related quality of life (OHRQoL)

3.5.2 The pain perception

3.5.2 Maximum Biting Force

  • Discussion: -This section was rewritten with the addition of more clarifications.

In addition,

  • we have edited the manuscript according to the other reviewers' comments, and the track changes evidence this.

    Thank you again for your precious comments and guidance.